# Simulation-Based Process Design for Asymmetric Single-Point Incremental Forming of Individual Titanium Alloy Hip Cup Prosthesis

**DOI:** 10.3390/ma15103442

**Published:** 2022-05-10

**Authors:** Sirine Frikha, Laurence Giraud-Moreau, Anas Bouguecha, Mohamed Haddar

**Affiliations:** 1Life Assesment of Structures, Materials, Mechanics and Integrated Systems (LASMIS), University of Technology of Troyes, 12 Rue Marie Curie, 10004 Troyes, France; laurence.moreau@utt.fr; 2Laboratory of Mechanics, Modeling and Production (LA2MP), National Engineers School of Sfax, Route de Soukra, Sfax 3038, Tunisia; anas.bouguecha@gmx.de (A.B.); mohamed.haddar@enis.tn (M.H.)

**Keywords:** incremental forming, finite element simulation, biomedical implants, titanium, wall angle

## Abstract

Advanced manufacturing techniques aimed at implants with high dependability, flexibility, and low manufacturing costs are crucial in meeting the growing demand for high-quality products such as biomedical implants. Incremental sheet forming is a promising flexible manufacturing approach for rapidly prototyping sheet metal components using low-cost tools. Titanium and its alloys are used to shape most biomedical implants because of their superior mechanical qualities, biocompatibility, low weight, and great structural strength. The poor formability of titanium sheets at room temperature, however, limits their widespread use. The goal of this research is to show that the gradual sheet formation of a titanium biomedical implant is possible. The possibility of creative and cost-effective concepts for the manufacture of such complicated shapes with significant wall angles is explored. A numerical simulation based on finite element modeling and a design process tailored for metal forming are used to complete the development. The mean of uniaxial tensile tests with a constant strain rate was used to study the flow behavior of the studied material. To forecast cracks, the obtained flow behavior was modeled using the behavior and failure models.

## 1. Introduction

Sheet-metal-forming industries are showing much interest in the single-point incremental forming (SPIF) method because of its part design flexibility, low tooling cost, and improved formability. A thin sheet metal is tightly pressed and progressively deformed into the required shape utilizing localized plastic deformation and a rigid forming tool [1]. A CNC machine tool can efficiently design a simple or complicated tool path, and any changes to the final shape may be included instantaneously by adjusting the tool’s motion [2].

Furthermore, a multistep tool path [3] method is frequently used to raise the inclination of a manufactured component’s wall angles. Because of its flexibility and lack of dies, the method is better suited to prototyping and manufacturing very complicated 3D shapes. The SPIF technique, however, cannot be used for the mass manufacture of industrial components due to its long production time. Nevertheless, the SPIF technique is well-suited to small-batch or customized manufacturing, and the fast prototyping of metallic products. The extensive use of progressively produced components may be seen in a variety of sectors, including automotive, aircraft, shipping, and biomedical. SPIF was utilized to create a variety of custom biomedical implants such as knee arthroplasty and cerebral implants with a complicated shape [4]. However, according to various studies, several parameters such as material qualities, tool design, and tool path impact the formability of incrementally produced products.

Individual prostheses are typically created using a reverse engineering process to fit the patient. Behrens et al. [5,6] established a method for producing these hip prosthesis cups or acetabulum (Figure 1) using traditional sheet-metal-forming procedures, including stamping and hydroforming. Cup individualization in this scenario necessitates the creation of customized equipment such as the die and punch that are unique to each patient. Traditional stamping procedures, on the other hand, are prohibitively expensive for small numbers or bespoke prototypes. ISF technology offers a novel way to reduce the cost of solving the problem in small-scale production. It introduces the cost-effective use of metallic sheets for small-batch manufacture without the use of expensive or specialized machines. As a result, the most appropriate approach for producing a personalized hip prosthetic cup is the ISF process.

It is not easy to find the right material and manufacturing technique to create a personalized implant with optimal biomechanical qualities. The literature contains a wealth of information on the use of a variety of biomaterials, ranging from metals to bioceramics and biopolymers, but titanium (Ti) and its alloys remain the most widely used solutions because they have a low Young’s modulus (ensuring uniform stress distribution between implant and surrounding bone), excellent mechanical properties, and, in terms of biocompatibility, the ability to promote osseointegration. However, owing to their high cost and poor formability at room temperature, the incremental forming of titanium has received little attention.

Stresses and thereby the material’s formability are affected by the shape of the incrementally created part. Geometrical mistakes are determined by many geometrical characteristics in the component, the kind of material used for forming, and the thickness of the material. Different scholars [7,8,9,10,11,12] attempted to characterize geometrical faults on the basis of geometry. Verbert et al. [9] employed a feature identification algorithm to sort characteristics into groups such planes, ribs, ruled, and freeform surfaces.

Additionally, the geometrical precision of components is affected by the forming angle; in high angled parts, underforming is common, while in shallow wall angled geometries, the overforming and bulging of the bottom is common. Due to severe sheet underforming, which can exert negative radial stress on the tool directed from the middle of the part towards the wall, a nonhorizontal base surface of the part may hamper the forming process. Various studies looked into various remedies to this sort of inaccuracy. Essa et al. [13] proposed that extending a tool path towards the center of the part can plastically deform the bulge and lower its height by using a tool-path optimization technique to lessen the bulging of the bottom part.

The SPIF method has been explored since the technology was initially conceived. To establish the boundaries of the SPIF process to produce a part, different approaches are utilized. Using thickness decrease as an indication, Saidi et al. [14,15,16] proved that the limiting forming angle for grade 2 titanium is 45${}^{{}^{\circ}}$ at ambient temperature. With the increased usage of aluminum alloys, titanium alloys, and advanced high-strength steels in the automotive and biomedical industries, a difficult issue arose owing to the abrupt development of fractures during forming trials.

The design of the SPIF process might be improved by the precise prediction of failure to avoid early material fracture of the product to be fabricated. Iseki et al. [17] analyzed strain components in the contact region and hypothesized that, once the strains reached the strain limit forecasted by the empirical fracture forming limit (FFL) under plane strain conditions, the maximal forming depth of the cone portion created by ISF could be projected. Haque and Yoon [18] presented stress-based FLCs to predict the formability of various materials to overcome the limitations of strain-based FLCs for SPIF.

Fracture prediction using FLCs or FFLs is simple and effective for process development and preliminary analysis. It cannot, however, give comprehensive knowledge of fracture initiation and progression during ISF. To understand damage initiation and progression in the SPIF process, indepth examinations of fracture behavior were conducted. Strain and stress distributions in the deformed region dictate the material’s deformation behavior and fracture. Because of its simplicity, theoretical fracture prediction combining basic ductile damage models depending on strain or stress analysis is widely used. Huang et al. [19] utilized the force equilibrium approach for an approximate estimate of stress distribution in the area of contact between forming part and ISF tool by ignoring circumferential force components and the friction effect. The highest forming angle attainable without fracture in the lateral direction was estimated using the Oyane damage criterion.

However, because contact conditions in ISF are complicated, simplistic analytical models cannot accurately predict the material’s stress and strain states during the ISF process. Many studies used FE modeling to track material deformation throughout the ISF process. However, the contact circumstances and progressive deformation of the material in standard FE models of the ISF process modeling generated using commercial software hinder analyzing damage progression in the process. Complex damage models were included in commercial FE modeling software to correctly capture fracture initiation and damage progression in the ISF process.

According to Besson and Jacques [20], depending on the scale utilized to evaluate the start and progression of the fracture, there are two types of ductile fracture models: micromechanical and phenomenological models. Micromechanical models, such as the Gurson model, are quasi models resulting from rigorous micromechanical research, whereas phenomenological models, such as the Lemaitre model, are mostly based on macroscopic factors. Researchers employed a variety of damage models in their FE simulation of the ISF process. Different damage models may only be appropriate for certain loading circumstances. In the SPIF procedure, Malhotra et al. [21] included a novel fracture model further into FE analysis program LS-DYNA to forecast failure in conic and funnel shapes. Guzmán et al. [22] examined three GTN damage model versions to better forecast SPIF damage progression. Results, however, did not match the trials, which essentially accepted the intricacy of the ISF process and the impact of work hardening on ISF fracture behavior.

Park et al. [23,24] included Barlat Yld91 and Hill48 yield theories into the Lou–Huh damage model to estimate anisotropic sheet-metal fracture strain. However, the prediction of fracture limit strain was minimally studied using a ductile fracture model that incorporates anisotropy of thin sheet metal.The adoption of an adequate plasticity model is crucial for accurately predicting failure strain during sheet metal forming. The Hill48 anisotropic plasticity model [25] was the first to replicate the anisotropic sheet metal yield locus. Yue et al. [26] developed a fully linked damage model with metal anisotropy and demonstrated the impact of material anisotropy on SPIF damage accumulation.

In this context, optimal process parameters can be specified to enable process or equipment design optimization and to prevent the early material failure of the forming piece throughout the production process by forecasting the material processing failure of a forming part in ISF. Furthermore, knowing the mechanism underlying material fracture would aid in avoiding any faults in the formed component, ensuring the ISF part’s quality and structural integrity.

The first aim of the present study is to predict the anisotropic behavior of commercially pure Grade 2 α titanium sheet metals in terms of inplane variation in properties and yield loci using the Hill48 yield model, and results were validated with the uniaxial tensile test. The necking and fracture limits of the sheet metals were estimated using a verified Johnson–Cook fracture model incorporating above calibrated yield theories. Theoretical failure limits were correspondingly derived through simulation tensile tests. The second aim was to develop a method to produce titanium hip cup prosthesis with a complex (asymmetric) shape and a wall angle, two-stage forming, combining single-point incremental forming and deep drawing. Moreover, the multistep approach was adopted, and a suitable tool path was generated. A tool was created that allows for the targeted profiles and simulated profiles to be compared to validate the proposed method. Eventually, the numerical part was enhanced.

## 2. Materials and Methods

### 2.1. Material and Experimental Characterization

The material used in this study was Grade 2 α titanium, which is commercially pure. Table 1 shows the chemical composition of the tested material, which is compared to the ASTM standard composition (in weight percentage) for this material. Titanium was purchased in the form of a 1 mm thin plate, which was subsequently sliced into samples using the ISO 6892-1 water-jet cutting procedure. Specimens were produced from material in the rolling (RD), normal (ND, orientation of 45 degrees with regard to RD), and transverse (TD) directions. Tensile tests were performed on these specimens using an Instron^®^ 4411 type testing equipment with a 5 kN capacity, as illustrated in Figure 2, at room temperature with a cross head speed of 1 mm/min to gather flow stress–strain data to analyze deformation flow behavior.

The study’s second batch of samples were used to look into the impact of stress triaxiality, mechanical behavior, and Johnson–Cook damage parameters. Smooth and notched samples were both subjected to tensile tests. Technical drawings of smooth and notched tensile test specimens oriented to rolling direction are shown in Figure 3. The notch radius of the notched specimens is represented by $R_{0}$. For the processing of specimens, three different notch radiuses were used. Specimen radii and gauge lengths are reported in Table 2.

To ensure result uniformity, each uniaxial monotonic tensile test was repeated four times for each case during the experiment. As a result, four repetitions for each of the six specimens were carried out for a total of 24 tensile tests. All trials were carried out with ISO standard 6892-1:2016 to dimension specimens.

A nondestructive image correlation technique was utilized to evaluate the deformations of a sample that is based on analysis of photos captured during mechanical testing. Digital cameras with high resolution captured a sequence of photos that tracked distortion over time. LAVISION^®^ software then interpreted the images. Two cameras (stereoscopic vision) were employed for the measurement of deformation fields in three dimensions. Force was measured using a load cell, and stress was calculated using the following standard strength equation:(1)$$\sigma =\frac{F}{A_{0}}$$
(2)$$\epsilon_{\mathrm{eng}}=\frac{\Delta \ell }{\ell_{0}}$$
(3)$$\begin{array}{cc} \sigma_{\mathrm{true}} & =\sigma_{\mathrm{eng}}\left(1+\epsilon_{\mathrm{eng}}\right);\\ \epsilon_{\mathrm{true}} & =\ln(1+\epsilon_{\mathrm{eng}}), \end{array}$$
where σ, stress (in MPa); *F*, force (in N); $A_{0}$, section (in mm${}^{2}$); $\epsilon_{\mathrm{eng}}$, engineer strain (unitless); Δℓ, displacement (in mm); $\ell_{0}$, initial length (in mm); $\sigma_{\mathrm{true}}$, real stress (in MPa); $\sigma_{\mathrm{eng}}$, engineer stress (in MPa); and $\epsilon_{\mathrm{true}}$, real strain (unitless).

Large deformations in the shaping process necessitate extrapolation. We chose two alternative models for this: Voce and Swift.


Swift (or Krupkowski) model:
(4)$$\begin{array}{cc} \sigma_{\mathrm{eq}} & =K{\left(\epsilon_{0}+\epsilon \right)}^{n} \end{array}$$
where $\epsilon_{0}$ and n define hardening constants. As a result, this work hardening law was tailored to the upper reaches of the cold deformation domain and can also be used for hot deformation.


The equation defines the identification of plastic anisotropy or Lankford’s coefficient:(5)$$\begin{array}{c} r_{\alpha }=\frac{\varphi_{w}}{\varphi_{th}} \end{array}$$
where $\varphi_{w}$, logarithmic strain in width direction; $\varphi_{th}$, logarithmic strain in thickness direction.

To calculate $r_{\alpha }$, it must be able to calculate both transverse plastic strain $\epsilon_{22}$ and thickness strain $\epsilon_{33}$, induced during a uniaxial tensile test. However, determining thickness deformation can be difficult. The plastic incompressibility hypothesis is explored as a solution to this problem.

Materials used in this study were rigid plastic and follow Hill’s yield criterion (1948), which is an extension of the von Mises’ criterion that takes into consideration plastic anisotropy. Hill48 parameters are presented in the equations below:(6)$$\begin{array}{c} \;\;\;\;\;\;G=\frac{2}{1+r_{0}}; \end{array}$$(7)$$\begin{array}{c} \;\;\;\;\;\;H=\frac{2r_{0}}{1+r_{0}}; \end{array}$$(8)$$\begin{array}{c} \;\;\;\;\;\;F=\frac{H}{r_{90}}; \end{array}$$(9)$$\begin{array}{c} L=M=N=\frac{\left(r_{90}+r_{0}\right)\left(2r_{45}+1\right)}{2r_{90}\left(1+r_{0}\right)}. \end{array}$$

The material is subjected to complex stresses during superplastic forming, which might result in defects (cracks, ruptures, etc.) within the part. Titanium, which is susceptible to cavitation [27], is prone to faults caused by ductile damage [28,29]. Once this topic is included in numerical simulations, it is possible to foresee material deterioration and avoid the occurrence of faults in the part. The damage of pure commercial Grade 2 titanium exposed to conditions similar to those found during superplastic formation is examined in this study. This paper proposes a Johnson–Cook type damage criterion.

**Johnson–Cook failure criterion**
The Johnson–Cook criterion is a phenomenological approach. According to Johnson and Cook, stress triaxiality ratio, strain rate, and temperature influence strain at break. The J–C rupture model is written as follows:(10)$$\begin{array}{c} \epsilon_{f}=\left(D_{1}+D_{2}exp\left(D_{3}\frac{P_{h}}{\sigma_{eq}}\right)\right)\left(1+D_{4}ln\frac{\dot{\epsilon_{p}}}{\dot{\epsilon_{0}}}\right)\left(1-D_{5}{\left(\frac{T-T_{a}}{T_{f}-T_{a}}\right)}^{m}\right) \end{array}$$
where $D_{1}$ to $D_{5}$ are constants of the damage model, $P_{h}$ is mean stress or hydrostatic pressure, and $\sigma_{eq}$ is equivalent stress [30]. A cumulative damage law is used to define an element’s damage, which can be linearly written as follows:(11)$$\begin{array}{c} D=\frac{\sum_{\Delta \epsilon }}{\epsilon_{f}} \end{array}$$
where Δϵ is the equivalent plastic strain increment, and $\epsilon_{f}$ It is the equivalent fracture strain at the current load, strain rate, and temperature circumstances. The material’s resistance diminishes during deformation as a result of the appearance of the fracture, and the constitutive relationship of stress for the evolution of damage can be described as follows:(12)$$\begin{array}{c} \sigma_{D}=\left(1-D\right)\sigma_{eq} \end{array}$$

$\sigma_{D}$ is damaged stress state, and *D* is damage parameter (0≤D<1) in Equation (Equation 11). Furthermore, the triaxiality of stresses [31,32,33,34] and equivalent stress may be calculated from the undamaged material while accounting for plastic behavior up to the development of the necking [30].

To acquire a better result, un-notched and notched samples were created using the water-jet cutting method in the first stage, and more than three samples were employed to obtain the flow curve. to manage experimental inaccuracy in terms of the material’s mechanical properties, as shown in Figure 4, a series of experiments involving un-notched and notched flat samples were simultaneously conducted at room temperature under a variety of quasistatic strain rate conditions to investigate the effect of triaxiality of constraints on the damage behavior of Grade 2 α-titanium.

For the identification of material properties and for the inputs of the material digital model, flow curves acquired from the tests were divided into elastic and plastic areas up to the necking. Furthermore, the material’s mechanical properties [30] were rigorously determined through experiments, as these properties are required to precisely reproduce the material’s behavior in real-world settings. Furthermore, predicted properties were included in commercial tools, and work hardening behavior was simulated using the isotropic work hardening model for stress triaxiality evaluation.

The estimation of the mechanical properties of the experiments was flawlessly completed, and it can be used to estimate the triaxiality of stresses without difficulty. Stress components such as $\sigma_{1}$, $\sigma_{2}$ and $\sigma_{3}$ are then determined using the average of the items in the sample that fail. To determine mean stress Ph and equivalent stress $\sigma_{eq}$, stress components were swapped in Equation (Equation 12).
(13)$$\begin{array}{c} \tilde{\sigma }=\frac{\left(\sigma_{1}+\sigma_{2}+\sigma_{3}\right)}{3*\sqrt{0.5*[{\left(\sigma_{1}-\sigma_{2}\right)}^{2}+{\left(\sigma_{2}-\sigma_{3}\right)}^{2}+{\left(\sigma_{1}-\sigma_{3}\right)}^{2}]}} \end{array}$$

The failure model equation can only be recast in terms of the influence of triaxiality of stresses concerning break strain by rearranging Equation (Equation 9) by disregarding the effects of strain rate and temperature:(14)$$\begin{array}{c} \epsilon_{f}=D_{1}+D_{2}exp\left(D_{3}\tilde{\sigma }\right) \end{array}$$

Equation (Equation 13) can be changed by substituting stress triaxialities and related ultimate strain values.

### 2.2. Forming Strategy

The proposed manufacturing method is a hybrid of two forming methods. The first is the deep-drawing method for producing conventional shapes. Deep drawing provides superior mechanical component characteristics while also reducing production time. It is a lengthy forming procedure. The second is the incremental forming procedure that is used to create unique and complex shapes for the cup outline that adheres to the bone.

The incremental sheet-metal-forming (ISF) technique is a viable alternative to traditional sheet-metal-forming processes such as hydroforming and deep drawing. It is a viable response to manufacturers’ desires for more flexible and cost-effective small-series and prototype solutions. For the prototyping and small-batch production of complicated sheet metal, incremental forming is ideal. Due to the need for product personalization for each patient, medical implants and prostheses are a primary potential field of application for the ISF process.

Forming geometric models with complex shapes, on the other hand, is still researched. Several experimental and numerical studies were conducted to investigate parameters impacting this process for various metals [14,35]. The incremental forming procedure used in the suggested approach allows for the creation of a custom prosthesis for each patient, rendering it more comfortable and ensuring greater performance.

We needed the cup’s geometry to simulate the process. Our method (Figure 5) is based on reverse engineering methodology with four steps:

Step 1: radiography of an individual pelvis; Step 2: definition of hip cup geometry; Step 3: the inside half of the cup is formed using a preforming process; Step 4: to manufacture the cup’s outside section, an incremental forming process was used.
Step 1 allows for the creation of a cloud of points that define the pelvic shape. This geometry was larger than the portion that had to be constructed.Step 2 specifies the cup’s geometry before it is constructed. Escobar et al. [5,6] presented a detailed description of the technique for obtaining this geometry. Initially, the fascia lunatic area is resected. The target region is fitted with a first sphere that is positioned so that the radius is the largest without entering the bone. As a result, the approximation sphere is better fitted. Then, to produce the outside contour, a second sphere is created around the first sphere. The inner component is produced by replacing the part’s core with a 20 mm diameter hemisphere. The final shape of the hip cup prosthesis is a titanium part with a large diameter of 100 mm, thickness of 1 mm, and a hemispherical region of 20 mm in the middle area with a wall angle of 70${}^{{}^{\circ}}$ to 75${}^{{}^{\circ}}$ in the outer zone (cf. Figure 6).During Step 3, the inner half of the cup is constructed using a preforming process. In the middle of the hip cup, a standardized geometry is defined. This procedure is perfectly suited to this situation.Step 4: The incremental forming procedure is utilized in Step 4 to build the cup’s outer section. At room temperature, titanium used to produce the hip cup had poor formability. Previous research demonstrated that forming direct parts with a wall angle larger than 40${}^{{}^{\circ}}$ at ambient temperature is unfeasible according to Saidi et al. [14]. The hip cup prosthesis is intended to be formed in multiple steps by increasing the wall angle.

**Figure 6 materials-15-03442-f006:**
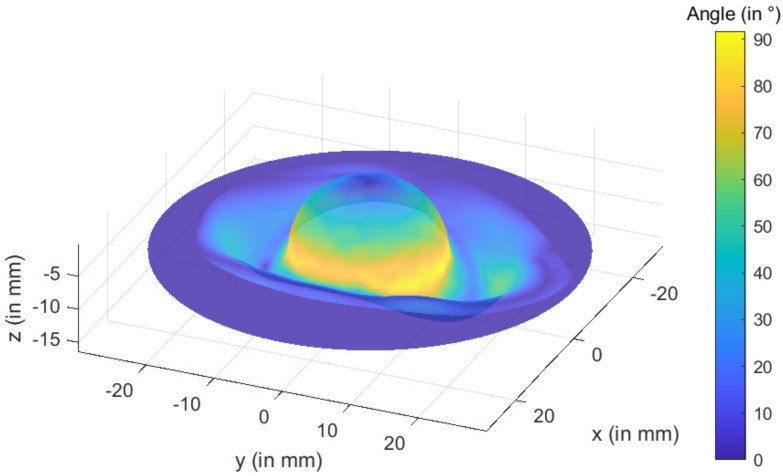
Analyzed part’s angle variation.

Formability, surface finish, thickness variation, processing time, and dimensional accuracy are all affected by the **tool path** in SPIF processes. The digital control provides the tool route (CNC). The tool follows a succession of contour lines with a vertical increment step size of Δz = 0.2 mm between them in a single forming step. The punch had a diameter of 2.5 mm. With a diameter of 100 mm and a thickness of 1 mm, the titanium alloy sheet was modeled into a round shape. The portion was created along the tool’s trajectory, as shown in Figure 7.

We began with a triangular depiction of the cup to specify the tool’s trajectory in the incremental forming phase. Indeed, the desired shape was modeled by a space mesh of triangles that defined the cup’s surface. We chose a spiral trajectory because of the cup’s shape, and the trajectory was specified by multiple successive passages to lessen the force applied to the part. The spiral followed the following equation in the horizontal plane, which was produced by the x and y axes:(15)$$\begin{array}{c} x\left(t\right)=\left(R_{i}+\left(R_{f}-R_{i}\right)\frac{t-t_{i}}{t_{f}-t_{i}}\right)cos\left(wt\right); \end{array}$$(16)$$\begin{array}{c} y\left(t\right)=\left(R_{i}+\left(R_{f}-R_{i}\right)\frac{t-t_{i}}{t_{f}-t_{i}}\right)sin\left(wt\right). \end{array}$$
where $R_{i}$ and $R_{f}$, spiral’s initial and final radii; $t_{i}$ and $t_{f}$, spiral’s initial and final times of passage; and ω, parameter that controls the spacing between spiral turns, i.e., angular speed of rotation.

Then, for each pair (x(t),y(t)) on the spiral, we designated the triangle that contained the point with coordinates $\left(x\left(t\right),y\left(t\right),z_{0}\right)$ and defined $z\left(t\right)=z_{0}$. Thus, path (x(t),y(t),z(t)) was defined as a tool path (cf. Figure 7).

In order to carry out several passages (cf. Figure 8), consecutive spirals were concatenated by alternating values of $R_{i}$ and $R_{f}$ and assigning progressive values to *z*(*t*):(17)$$\begin{array}{c} z\left(t\right)=\frac{i}{N}z_{0} \end{array}$$
where *N*, number of total passages; *i*, index of current passage (1≤i≤N).

### 2.3. Numerical Simulation



**Preforming**
To model the stamping of the inner portion, a numerical simulation of the sheet metal stamping process was first performed. Figure 9 depicts the punch and die. They are supposed to be rigid, and rigid surfaces are used to model them.During this simulation, the sheet was first placed on the die by being merely subjected to gravity, formed using the punch action, and released. Interaction parameters between contact zone and sheet were defined using the Coulomb friction model. Quadrilateral components with reduced integration S4R that were chosen to mesh the part were used.
**Incremental forming**
The SPIF process was numerically simulated using ABAQUS FE software. The dynamic displacement explicit solver was used to model the finite elastoviscoplastic deformation of a thin titanium sheet with contact friction. Large-strain formulation and linear four-node coupled stress/displacement shell elements (S4RT) with reduced integration were employed. The sheet’s mesh was partitioned into three zone divisions to minimize computational calculations (cf. Figure 10):sweep mesh (finite elements with a size of 10 × 0.5 mm) in the clamping zone;circular fine mesh (finite elements with a size of 0.5 × 0.5 mm) in the useful zone;fine mesh (finite elements with a size of 0.25 × 0.25 mm) in the critical zone.The part is created in the second stage using the tool trajectory shown in Figure 7. The forming tool follows a specified tool path to form the sheet metal in a series of incremental steps until the desired depth is reached. For incremental forming, tools (die and punch) are regarded to be rigid analytical surfaces. The punch is transformed into a diameter–diameter ball (dp = 0.4 mm) and is inflicted by displacement. The blank holder, modeled by rigid surfaces, is the stationary portion. However, the die on which the sheet sits was fixed according to the (x,y) plane and followed the tool’s movement in z, and surface-to-surface contact between punch and sheet. The sheet was tightly secured to the fixture (the blank holder) and was experimentally supported on it. The sides of the initial sheet were set in all directions from a numerical standpoint and for each simulation (cf. Figure 10).The chosen tool path for this study was composed of rotational movements in the horizontal plane combined with small steps in the vertical direction (ΔZ = 0.2 mm) after every round. The feed speed of $V_{f}=600$ mm/min was chosen.One of the most critical concerns for accurately simulating the incremental forming process is modeling the interaction between tool and sheet. Coulomb’s friction model, which is described as follows, defines the interaction qualities between sheet and contact region with the spherical punch:
(18)f=μN.
where (*f*) is frictional shear stress, μ=0.1 [36] is the coefficient of friction, and *N* is the normal contact pressure.The viscoplastic material of Grade 2 α titanium was defined using the experimental tensile test data presented in Section 3.1. The Swift model was selected to extrapolate data even though shaping processes need huge plastic deformations. The Hill48 yield model was also incorporated. The Johnson–Cook damage model was chosen because of its ability to take into account the triaxiality in the material modeling during numerical simulation to predict failures.


## 3. Results and Discussion

### 3.1. Experimental Results

Figure 11 shows a stress–strain curve with a clear yield point and high ductility of the smooth specimen.

Overall, obtained results were consistent according to the Grade 2 α titanium abacus. We could also compare the different standard deviations (cf. Table 3) of each variable for each rolling direction. Table 3 demonstrates the consistency of the experiments. At 90${}^{{}^{\circ}}$, there was a high difference in the value of tensile strength at 45${}^{{}^{\circ}}$ linked to the various uncertainties mentioned above. The superposition of flow curves in three directions is depicted in the diagram (cf. Figure 12).

Figure 13 shows a comparison between experimental data and the Swift model. Voce’s model gave an error of $1.07\times 10^{-1}$, whereas Swift’s model gave an error of $8.27\times 10^{-3}$. As a result, Swift’s model was employed. The values of Swift model parameters are shown in Table 4.

The change in anisotropy coefficient during the room-temperature strain hardening test for the rolling direction and a strain rate of 1 mm/min was illustrated. The plot of $\epsilon_{22}$ as a function of $\epsilon_{33}$ (obtained by considering plastic incompressibility) shows that the plastic anisotropy coefficient was not very sensitive to the level of plastic deformation. All values of identified anisotropy coefficients are given in Table 5. These values are in agreement with those provided by Odenberger et al. [37] for Grade 2 α titanium at the same temperature.

The calculated anisotropic coefficients at ambient temperature were between 4 and 5.5 (Table 5). This means that the plastic strains in the transverse direction were greater than those in the thickness. The material thus tended to elongate rather than thin out.

With the values of Lankford coefficients $r_{0},r_{45},r_{95}$, it is possible to quickly calculate coefficients of the anisotropic criterion of Hill48 as shown in Table 6:

The plot of relation $\epsilon_{\tilde{f}}\tilde{\sigma }$ was developed from the triaxiality curve, as shown in Figure 14; model parameters D1, D2 and D3 were calculated and are shown in Table 7.

### 3.2. Numerical Results



**Preforming**
Figure 15 shows the visual inspection of standardized human hip cup’s generated components. At room temperature, simulating the preforming process yielded satisfactory results, with a 20% reduction in blank thickness, which is tolerable thickness. Figure 15 also depicts homogeneous thickness distribution.Regarding the forming process at room temperature, Figure 16 displays the isovalues of the scalar damage parameter described by stiffness degradation SDEG. SDEG=1.0 indicates that the appropriate elements failed, and a fracture formed. Damage parameter SDEG was substantially lower than zero in these data, confirming the earlier result that the inner section forms at Ti = 20 C.We estimated the root mean square error (MSE) of altitude z of each point on the X–Y plane to compare theoretical and numerical profiles. The largest distance between two orthogonal locations on the surface is termed the error. As seen in Figure 17, the error was 2.38, which is small when compared to the sheet’s diameter, indicating excellent conformance.
**Incremental forming**



**Figure 15 materials-15-03442-f015:**
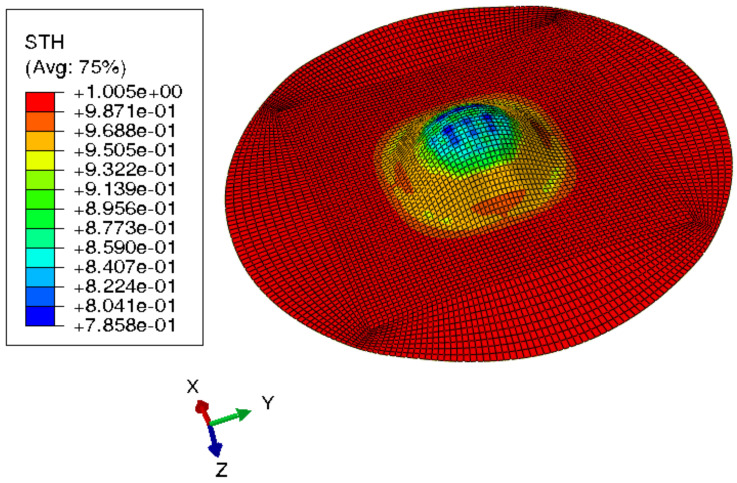
Distribution of shell thickness decreasing at ambient temperature: preforming process.

**Figure 16 materials-15-03442-f016:**
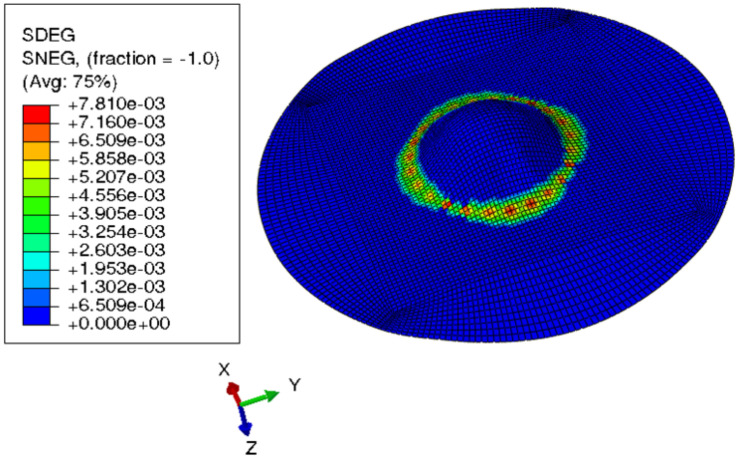
3D iso-values of stiffness degradation determined by finite-element simulation for preforming process.

**Figure 17 materials-15-03442-f017:**
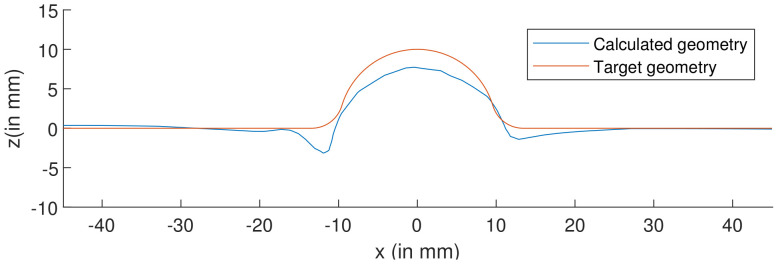
2D Comparison between target and simulated geometry of inner part.

#### 3.2.1. One-Step Forming

Because the SDEG value in the contact zone between punch and sheet was extremely near to 1, as seen in Figure 18, critical damage emerged at room temperature. The damage variable increased as predicted in these settings. These findings were given at the same time as the part’s maximal deterioration at room temperature. The item was fully ruined in a short time at Ti = 20 ${}^{{}^{\circ}}$C, as predicted, because titanium has very poor formability at ambient temperature.

Figure 19 depicts the titanium blank’s thickness reduction as a result of simulating the one-step incremental forming process at room temperature. According to modeling results, one-step incremental forming at room temperature would result in 80% reduction in blank thickness, which is critical and a tear. It is difficult to finish this method without a cracked blank at the end due to its thinning by more than 25%, which is considered to be a limit to avoid the risk of cracking. It confirmed the prior finding that the piece does not form at Ti = 20 ${}^{{}^{\circ}}$C.

#### 3.2.2. Multistep Forming

Because damage parameter SDEG was substantially less than 1, Figure 20 demonstrates that the component was not harmed. According to the models, multistep incremental shaping for titanium blanks might reduce blank thickness by roughly 25%. Blank thickness is improved by simulating multistep incremental forming at the same temperature. Figure 21 shows a titanium blank and its damage parameter SDEG as a result of modeling multistep incremental forming at room temperature.

Modeling one- and multistep incremental forming processes on a titanium blank indicated that multistep incremental forming at room temperature is the most viable method for creating the prosthetic acetabulum. As a result, there was one more step to take.

Taking the damaging effect into consideration, the greatest harm occurred in one-step formation. Because the plasticity of titanium is better in multistep incremental forming operations than that in single-step procedures, ductility was substantially improved, and yield stress was cut in half. These results are relatively small compared to results obtained by Duflou et al. [38]. Steps 2 to 4 should be chosen in the researched range to produce improved formability in terms of forming penetration and geometrical precision.

According to the models, multistep incremental shaping for titanium blanks might reduce the thickness of the blank by roughly 26%. Blank thickness is improved by simulating multistep incremental forming at the same temperature. Figure 22 depicts the titanium blank and its thickness decreasing as a result of modeling multi-step incremental forming at room temperature, confirming prior findings.

This conclusion is supported by the experiences of a number of authors, including A. Bouguecha et al. [5,6,39].

Simulation results were compared with the specified hip cup target. For this comparison, Catia V5^®^ (Dassault Systems, France) was used to construct the geometry of a 3D scan of the target component. The digitized geometry was compared with simulation results to discover the best fit. In this experiment, MATLAB^®^ was used. Results of best fit and deviation analysis are shown in the diagram. Dimensional deviation analysis (cf. Figure 23) showed minimal variation of 3.6 mm in the contour and standard deviation of 2.52 mm across the board. As a result, simulation and theoretical components highly agreed.

## 4. Conclusions

An experimental and numerical study on personalized hip cup prosthesis accuracy enhancement in single-point incremental sheet-forming technique SPIF of titanium sheets was described in this work. Key benefits of this configuration are its low cost and ease of use, which allows for it to be used throughout the industry. At room temperature, experimental tests were carried out to determine the properties of the titanium sheet.

This study’s findings are summarized as follows:The SPIF of the acetabular utilizing titanium sheets is a realistic technique that demonstrates the potential for real-world medical use.Multistep manufacturing improves geometry accuracy.Preliminary findings are promising, and the procedure appears to be suitable for the installation of a hip prosthesis.To optimize these parameters for experimental investigation, more work is needed to examine the influence of the majority of process factors on component formability.

The potential of using SPIF technology to produce a metal acetabular of a hip prosthesis was statistically examined in the presented study.

The importance of the utilized numerical prediction and behavioral model was demonstrated by comparing experimental and numerical simulation findings on the basis of the thickness distribution, damage prediction, and final profile shape of the customized hip cup prosthesis. Profile differences between numerical and experimental findings were less than 4%. The suggested FE model may accurately predict experimental outcomes of customized titanium hip cup prostheses.

Thickness reduction findings for the customized hip cup prosthesis (an asymmetric shape with a nonhorizontal base) revealed that the ambient forming of titanium is acceptable. This guarantees that the material’s ductility is improved by combining two processes: conventional and incremental shaping. As a result, heating can reduce inaccuracies between desired and computed geometry. To reduce these errors, in future work, we will focus on improving the FE simulation tool, optimizing the tool path, and testing warm incremental sheet forming. In this work, only the numerical part was examined, and the experimental part will be carried out in future works.

## Figures and Tables

**Figure 1 materials-15-03442-f001:**
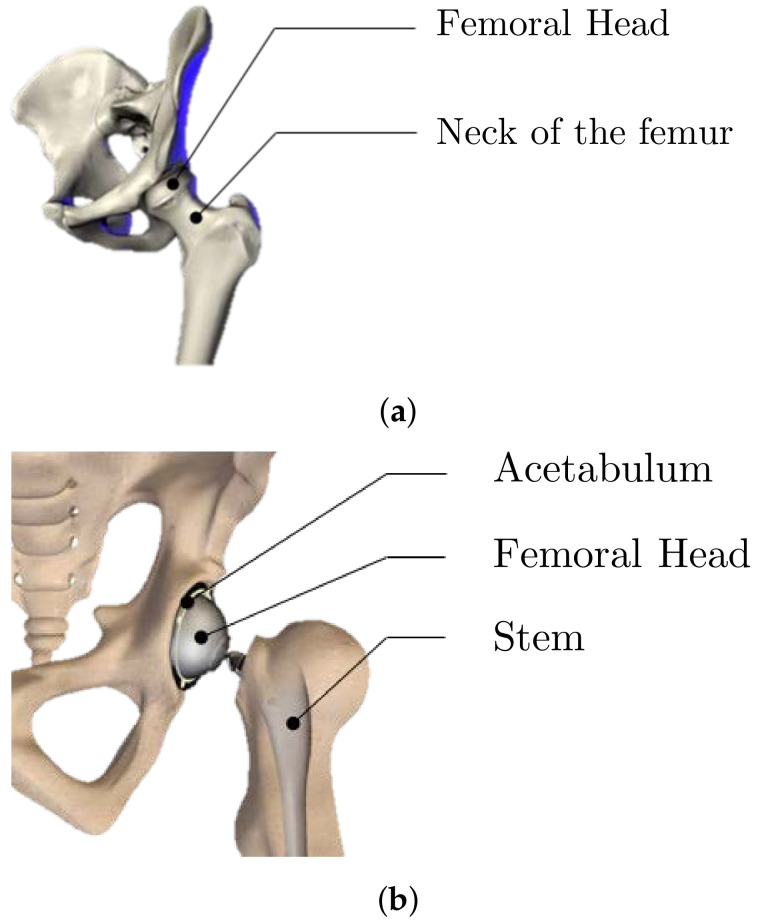
Pelvic bone (**a**) before and (**b**) after total hip arthroplasty.

**Figure 2 materials-15-03442-f002:**
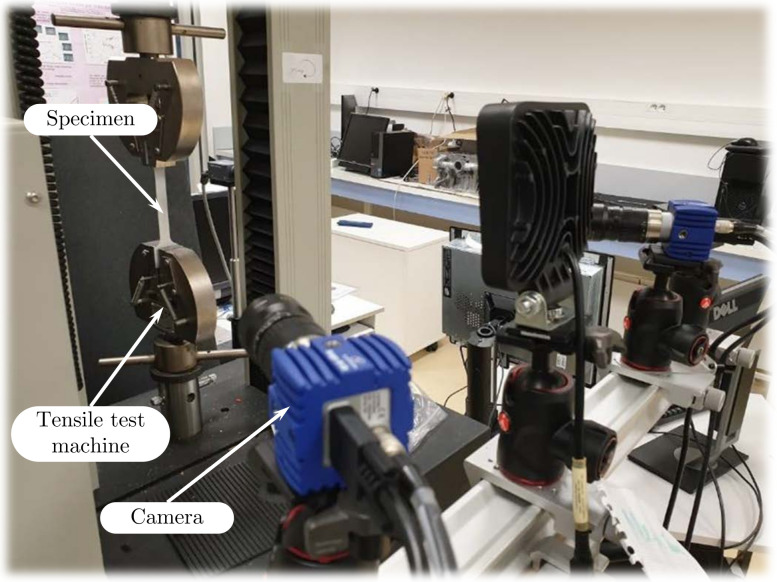
Experimental setup.

**Figure 3 materials-15-03442-f003:**
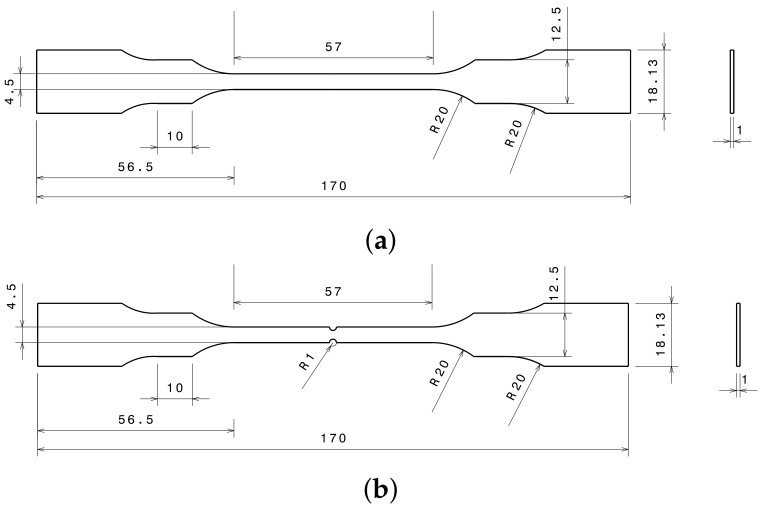
CAD design of tensile test specimens. (**a**) Un-notched specimen. (**b**) Notched specimen.

**Figure 4 materials-15-03442-f004:**
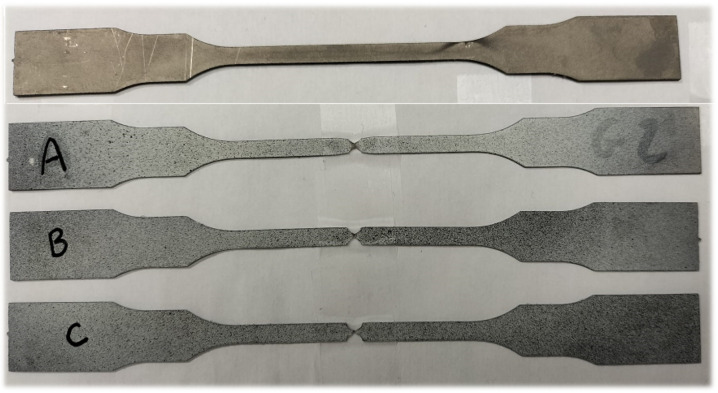
Notched (A: R = 0.5 mm; B: R = 1 mm; C: R = 1.5 mm) and un-notched specimens used in tensile tests.

**Figure 5 materials-15-03442-f005:**
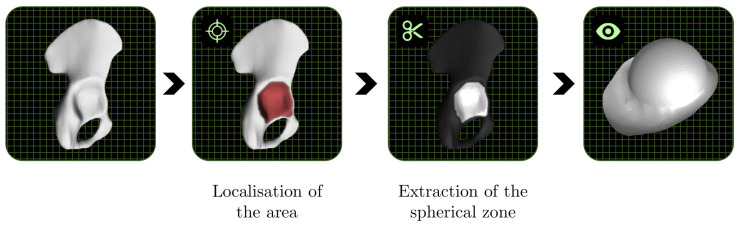
CAD creation of hip cup.

**Figure 7 materials-15-03442-f007:**
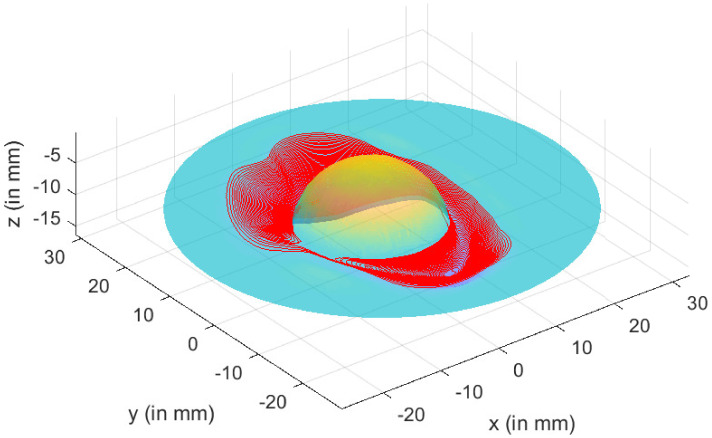
Tool trajectory—one step.

**Figure 8 materials-15-03442-f008:**
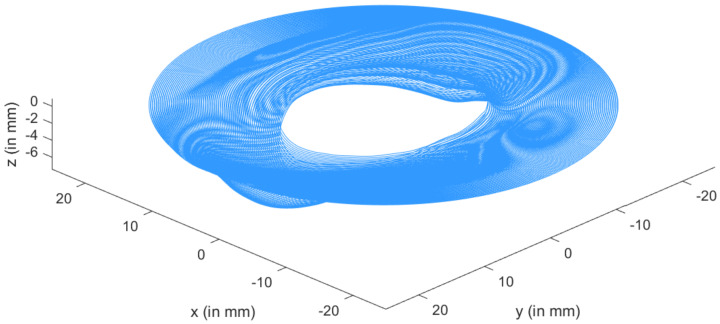
Tool trajectory—three steps.

**Figure 9 materials-15-03442-f009:**
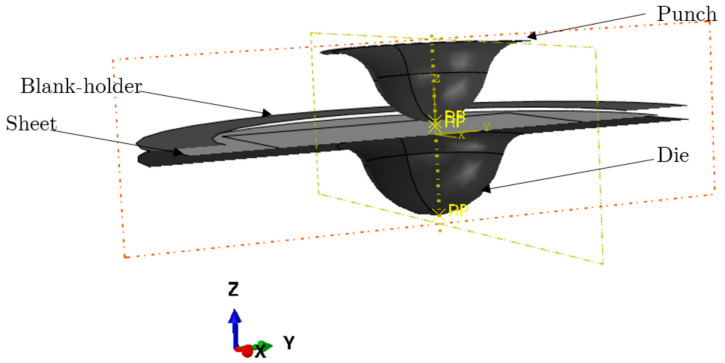
FE model—deep drawing.

**Figure 10 materials-15-03442-f010:**
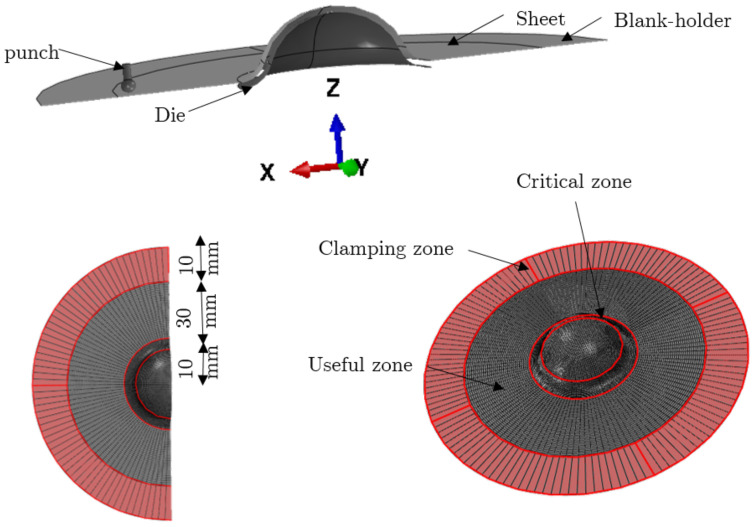
3D CAD model, FE meshing, and partition of initial sheet.

**Figure 11 materials-15-03442-f011:**
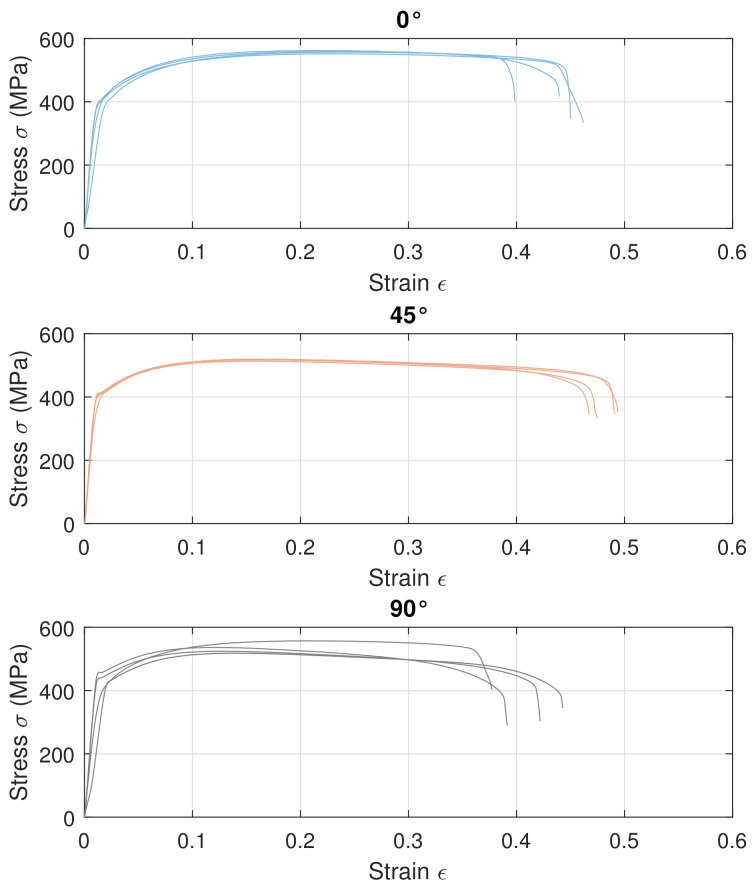
Conventional curve of Grade 2 titanium.

**Figure 12 materials-15-03442-f012:**
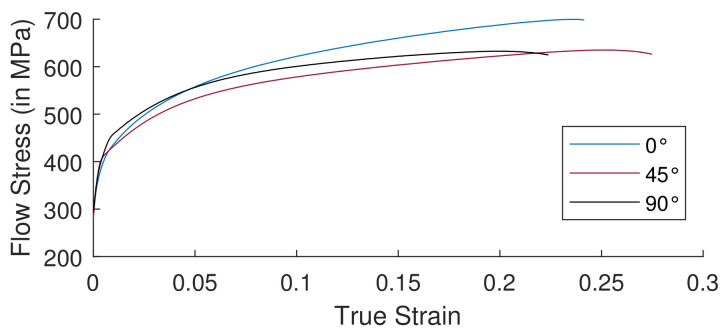
Flow curve of Grade 2 titanium for 3 directions.

**Figure 13 materials-15-03442-f013:**
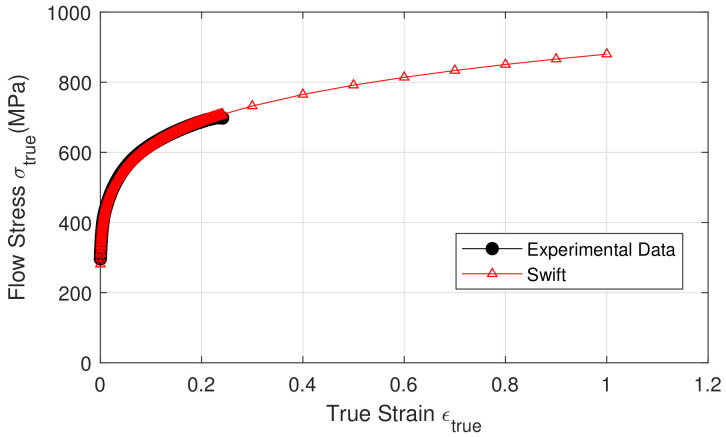
Comparison between experimental data and Swift model.

**Figure 14 materials-15-03442-f014:**
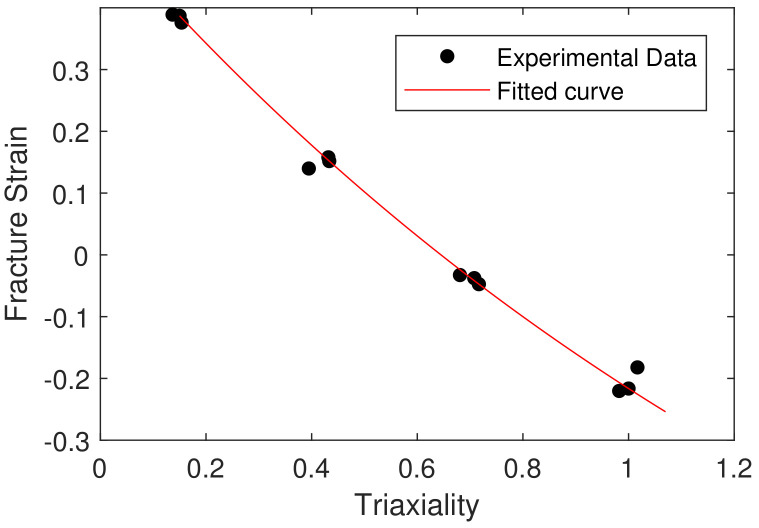
Fracture strain as function of stress triaxiality.

**Figure 18 materials-15-03442-f018:**
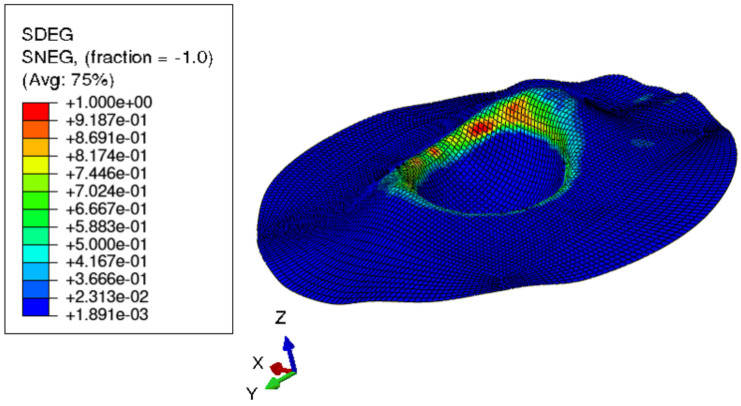
3D isovalues of stiffness degradation determined by finite-element simulation for one-step forming.

**Figure 19 materials-15-03442-f019:**
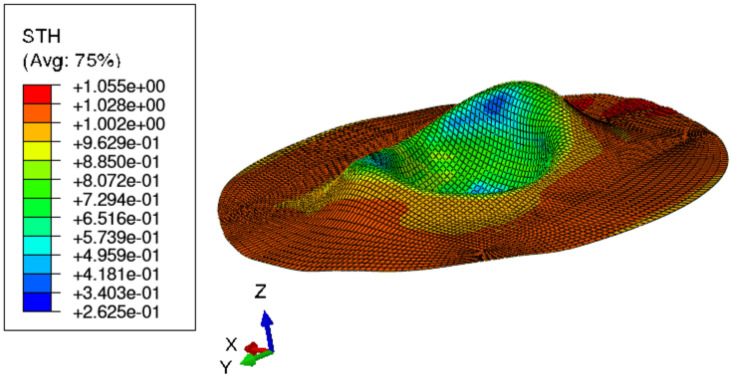
Distribution of shell thickness decreasing at ambient temperature: one-step forming.

**Figure 20 materials-15-03442-f020:**
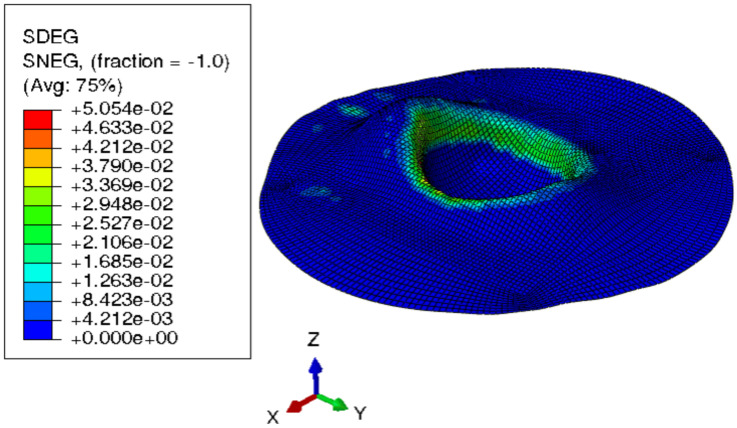
3D isovalues of stiffness degradation determined by finite-element simulation for intermediate step forming.

**Figure 21 materials-15-03442-f021:**
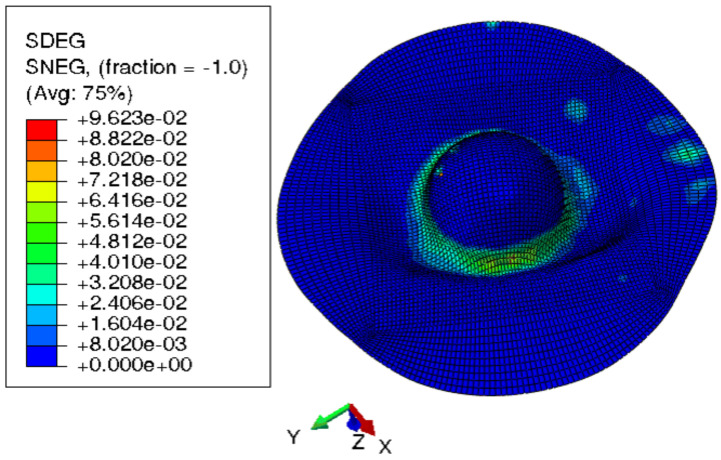
3D isovalues of stiffness degradation determined by finite-element simulation for multistep forming.

**Figure 22 materials-15-03442-f022:**
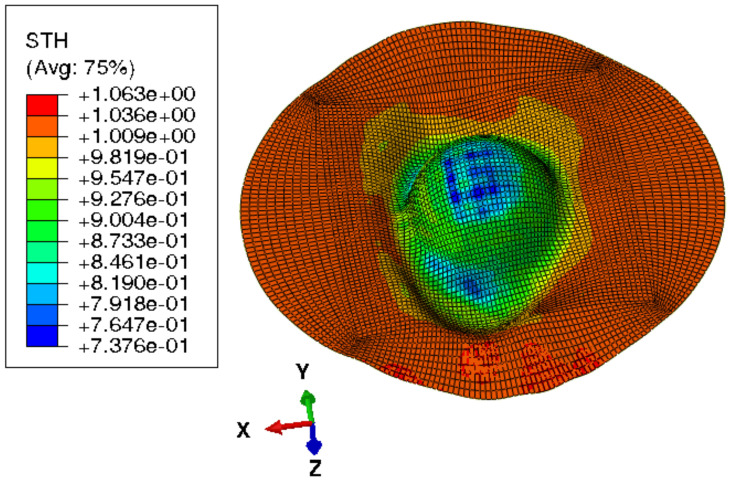
Distribution of shell thickness decreasing at ambient temperature: multistep forming.

**Figure 23 materials-15-03442-f023:**
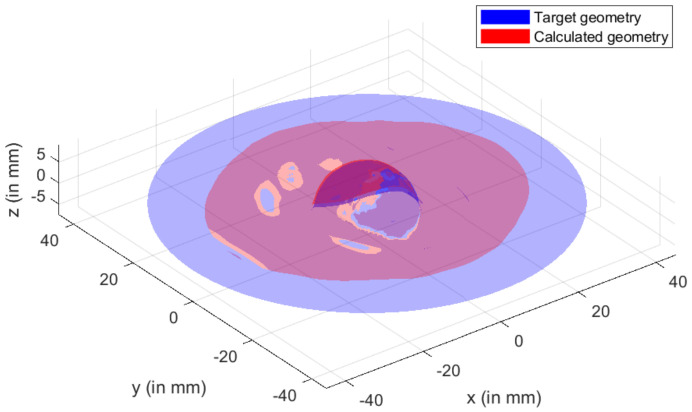
3D Comparison between target and simulated geometry of hip cup prosthesis.

**Table 1 materials-15-03442-t001:** Comparison of ASTM standard and chemical composition of Grade 2 α-titanium.

Chemical Composition (wt%)	Used Sample	ASTM F67
**Titanium**	49.7%	32.5%
**Iron**	20%	30%
**Oxygen**	18%	25%
**Carbon**	6%	8%
**Nitrogen**	5%	3%
**Hydrogen**	1.3%	1.5%

**Table 2 materials-15-03442-t002:** Notch radius and gauge length of the studied material.

Material	Notch Radius R (in mm)	Gauge Length (in mm)
Rolling Direction	0	57
	0.5	56
	1	55
	1.5	54

**Table 3 materials-15-03442-t003:** Mechanical properties of Grade 2 titanium for 3 rolling directions obtained from 4 tensile tests (mean ± standard deviation).

Rolling Angle	E (GPa)	Rm (MPa)	Re (MPa)	A
**0${}^{{}^{\circ}}$**	111.92 ± 0.29	288.50 ± 4.53	422.05 ± 5.00	32.25% ± 1.71%
**45${}^{{}^{\circ}}$**	111.90 ± 0.18	288.75 ± 41.0	368.45 ± 6.29	35.90% ± 0.84%
**90${}^{{}^{\circ}}$**	112.02 ± 0.17	325.25 ± 51.6	373.12 ± 5.68	35.25% ± 2.50%

**Table 4 materials-15-03442-t004:** Swift law.

Material Characteristics	K[MPa]	$\epsilon_{0}$	n
**T40**	**880**	**0.0578**	**0.153**

**Table 5 materials-15-03442-t005:** Anisotropy coefficients for Grade 2 α titanium.

Orientation (${}^{{}^{\circ}}$)	r
0	5.43
45	4.57
90	4.1

**Table 6 materials-15-03442-t006:** Coefficients of Hill48’s anisotropic criterion for Grade 2 α titanium.

Hill48 Coefficients	Values
G	0.31
H	1.68
F	0.41
L = M = N	1.83

**Table 7 materials-15-03442-t007:** Johnson–Cook failure parameters for Grade 2 α titanium.

Material Parameters of Grade 2 α Titanium	Values
$D_{1}$	−1.157
$D_{2}$	1.685
$D_{3}$	−0.583

## Data Availability

Not applicable.

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
