# Peer review of "Simulation-Based Process Design for Asymmetric Single-Point Incremental Forming of Individual Titanium Alloy Hip Cup Prosthesis"

_materials, 2022, doi:10.3390/ma15103442_

Round 1

Reviewer 1 Report

In this work, a Simulation-based process design for the asymmetric single point incremental forming was conducted to produce individual hip cup prosthesis made of titanium alloy. This paper is easy to follow, has representable figures, and the references cited appear to be in proper order. However, some issues are mentioned here which need to explain by the authors.

  1. The novelty of the present work should be highlighted in the introduction section. It should be emphasized so that the paper looks like different from the similar kind of previous works.
  2. Please incorporate a few more literature in incremental forming in the introduction portion to show the recent works and trends of the SPIF processes. Especially there is a tendency to incorporate the anisotropy model for fracture estimation during the ISF process. Recently, there is a tendency to couple the anisotropy model with the fracture model. This tendency of research should be reflected in the literature study. These discussions will be helpful to understand the recent advancement on the ISF. From a quick search, I can recommend some of the recent works on the field of SPIF.
  • Mohammadi, A., Vanhove, H., Van Bael, A. and Duflou, J.R., 2016. Towards accuracy improvement in single point incremental forming of shallow parts formed under laser assisted conditions.  International Journal of Material Forming9(3), pp.339-351.
  • Basak, S., Prasad, K.S., Sidpara, A.M. and Panda, S.K., 2019. Single point incremental forming of AA6061 thin sheet: calibration of ductile fracture models incorporating anisotropy and post forming analyses. International Journal of Material Forming12(4), pp.623-642.
  • Khazaali, H. and Fereshteh-Saniee, F., 2019. Process parameter enhancement for incremental forming of titanium Ti–6Al–4V truncated cone with varying wall angle at elevated temperatures. International Journal of Precision Engineering and Manufacturing, 20(5), pp.769-776.
  • Basak, S. and Panda, S.K., 2019. Failure strains of anisotropic thin sheet metals: Experimental evaluation and theoretical prediction. International Journal of Mechanical Sciences, 151, pp.356-374.
  • Esmaeilpour, R., Kim, H., Asgharzadeh, A., Nazari Tiji, S.A., Pourboghrat, F., Banu, M., Bansal, A. and Taub, A., 2021. Experimental validation of the simulation of single-point incremental forming of AA7075 sheet with Yld2004-18P yield function calibrated with crystal plasticity model. The International Journal of Advanced Manufacturing Technology, 113(7), pp.2031-2047.
  1. What is the normal direction (ND) of the tensile test? Do you want to refer it as 45deg. orientation with respect to RD of the sheet material? If it is so, then please refer it to the diagonal direction. Clarify what you are actually indicating by the ND of the tensile test?
  2. From which sheet orientations the specimen of Fig.5a and Fig.5b is cut? Please clarify in the text. Are the specimen dimension mentioned in Fig.5a and 5b are from any standard? Please mention.
  3. Fig.16 the fitted curve is as per the JC damage curve or it is just any random curve fitted with the experimental data point? Clarify. As many set of experiments has been conducted to get the experimental data. Hence, it is advisable to plot all the experimental data in Fig. 16.
  4. Please discuss more about FE modeling of the ISF process. Which FE software has been used? Discuss about the meshing strategy, optimization of the mesh size especially at the deforming region, material modeling, etc. So that this information help readers to understand more about the process modeling in FE software. Also, indicate the time integration scheme and element formulation used in the present FE modeling. Is any type of re-meshing technique has been used? As there is huge plastic deformation during the process, has a new mesh generation scheme been used in the study to model the huge plastic deformation region? Without this information, the study of numerical results is kind of incomplete.
  5. There are certain differences in the target geometry and calculated geometry. How to overcome that differences? Suggest some method in the manuscript for obtaining exactly target geometry with close tolerance values.

Reviewer 2 Report

The authors should rewrite the introduction regarding the novelty and their works located in which place of state of art. The recently research work related with their work should be discuss in the detail. Their work are different with the original which should be described  and discussed in the detail. Why this work presented in this way. 

By the way the scientific graph must be redrawn with high resolution and much more sharpness, clearly presented in the best way.

Finally, the conclusion must be improved the presented way, more appropriate presented and discussed.

Good luck with your research work

Reviewer 3 Report

Dear Authors
The publication presented for review is very interesting. However, during careful analysis, several aspects appeared that require improvement / clarification
1.Introduction
Introduction is too long. Basically, the text should be shortened while leaving the most important information
2. Related work
The information contained in this point should be included in the point Intoduction after its significant abbreviation
Figure 2 does not reflect the essence of the ISF process
3.1. Material and experimental characterization
Can the material used be compared in terms of chemical composition with the material according to the standard? The difference in titanium content is as high as 17.2% and for iron 10%
Figure 3 is redundant
Why are the words ASTM and Instron® 4411 in bold face?
3.2. Forming Strategy
Figure 7 needs improvement
4.Results
Unfortunately, there is no single result of experimental research that could prove that the proposed theoretical solutions and computer simulations coincide with the real product.

Reviewer 4 Report

The abstract is concise. The manuscript features the following chapters: 1. Introduction, 2. Related Work, 3. Materials and Methods, 4. Results and Discussion, 5. Conclusions. The paper is very interesting and well structured, but only some suggestions are recommended to increase the quality of the manuscript.

1- Figure 7 doesn’t present high quality/resolution.

2- In my opinion, the definition of some variables / abbreviations could appear at the end of the tables.

3- To better explain the experimental setup, the authors should clarify whether there are any limitations during the experimental procedure.

4- Please give information about instrumentation calibration / accuracy / test cell velocity (why the adopted value?)…for the used tensile tests. And about the experiments, was the repeatability / consistency verified?

5-No reference to mesh discretization convergence studies, (element shape, DOF).

6-The value of coefficient of friction.

7-The authors use the Johnson-Cook model as the failure criterion to simulate the mechanical behavior during the plastic formation. What is your opinion relating with the Cowper-Symonds model? And why you don’t choose this model?

8-In conclusions, future developments need to be included.

Reviewer 5 Report

The introduction is very long and extensive in aspects that are of little relevance. The scientific challenge and the contribution of the research should be more focused and clearer.

I don't really see a relevant interest in performing numerical simulations on parts that can be formed relatively simply and quickly. In fact, this is one of the advantages of this technique. Moreover, the proposal that has been made does not show any kind of contrast and the quality of this contrast is unknown. Nor are all the boundary conditions, simulation variables, software used, etc. defined.  The authors should remember that one of the aspects of any scientific work is the reproduction of the tests by other groups. Would it not have been much more interesting to be able to compare the numerical simulation with the real test? 

Previous experience is not cited and there is a lack of a broader and more detailed discussion to put the results obtained in context with previous experience.

There is a lack of recommendations that would have given quality to the work. For example, the most appropriate values for Ø and type of tool, feed rate, z-step, drawing strategy, geometry of the sheet, way of fixing the sheet, real variation of the sheet thickness, etc.  A real design of experiments is missing.

The conclusions of the work are known and to be expected.

In my opinion, the research is incomplete, the results are not very robust and do not represent a relevant contribution to this scientific field.

Round 2

Reviewer 1 Report

accepted

Reviewer 2 Report

All requirements have been improved and developed that is enough for publishing in Journal of Materials.

Reviewer 3 Report

Dear Authors,
Thanks for the your responses. I accept all of them, except one. Why are the simulation results compared to those presented in Behrens, B .; Escobar, S .; ... and experimental research has not been carried out by the authors of the publication?

Reviewer 5 Report

Se han subsanado algunas de las carencias, aunque sigo creyendo que el valor de este trabajo sin contraste práctico es de poca relevancia. 
